# Anti-GRP-R monoclonal antibody antitumor therapy against neuroblastoma

Jingbo Qiao[1], Junquan Liu[2], Jillian C. Jacobson[1], Rachael A. Clark[1], Sora Lee[1], Li Liu[3], Zhiqiang An[2], Ningyan Zhang[2], Dai H. Chung[1,4]*

1 Department of Surgery, UT Southwestern Medical Center, Dallas, Texas, United States of America, 2 Texas Therapeutics Institute, Brown Foundation Institute of Molecular Medicine, McGovern Medical School, The University of Texas Health Science Center at Houston, Houston, Texas, United States of America, 3 Department of Radiology, UT Southwestern Medical Center, Dallas, Texas, United States of America, 4 Department of Surgery, Children's Health, Dallas, Texas, United States of America

* dai.chung@utsouthwestern.edu

**Data Availability Statement:** All relevant data are within the paper and its Supporting information files.

**Funding:** This work was supported by a grant from the National Institutes of Health (R01 DK61470).

## Abstract

Standard treatment for patients with high-risk neuroblastoma remains multimodal therapy including chemoradiation, surgical resection, and autologous stem cell rescue. Immunotherapy has demonstrated success in treating many types of cancers; however, its use in pediatric solid tumors has been limited by low tumor mutation burdens. Gastrin-releasing peptide receptor (GRP-R) is overexpressed in numerous malignancies, including poorly-differentiated neuroblastoma. Monoclonal antibodies (mAbs) to GRP-R have yet to be developed but could serve as a potential novel immunotherapy. This preclinical study aims to evaluate the efficacy of a novel GRP-R mAb immunotherapy against neuroblastoma. We established four candidate anti-GRP-R mAbs by screening a single-chain variable fragment (scFv) library. GRP-R mAb-1 demonstrated the highest efficacy with the lowest $EC_{50}$ at 4.607 ng/ml against GRP-R expressing neuroblastoma cells, blocked the GRP-ligand activation of GRP-R and its downstream PI3K/AKT signaling. This resulted in functional inhibition of cell proliferation and anchorage-independent growth, indicating that mAb-1 has an antagonist inhibitory role on GRP-R. To examine the antibody-dependent cellular cytotoxicity (ADCC) of GRP-R mAb-1 on neuroblastoma, we co-cultured neuroblastoma cells with natural killer (NK) cells versus GRP-R mAb-1 treatment alone. GRP-R mAb-1 mediated ADCC effects on neuroblastoma cells and induced release of IFNγ by NK cells under co-culture conditions *in vitro*. The cytotoxic effects of mAb-1 were confirmed with the secretion of cytotoxic granzyme B from NK cells and the reduction of mitotic tumor cells *in vivo* using a murine tumor xenograft model. In summary, GRP-R mAb-1 demonstrated efficacious anti-tumor effects on neuroblastoma cells in preclinical models. Importantly, GRP-R mAb-1 may be an efficacious, novel immunotherapy in the treatment of high-risk neuroblastoma patients.

## Introduction

Neuroblastoma is the most common extracranial solid tumor of early childhood [1, 2]. High-risk neuroblastoma is designated based on the demonstration of clinically advanced stage or

This work was supported in part by the Welch Foundation grant AU-0042-20030616 (ZA), and Cancer Prevention and Research Institute of Texas (CPRIT) Grants RP150230, RP150551, and RP190561 (ZA). The funders had no role in study design, data collection and analysis, decision to publish, or preparation of the manuscript.

**Competing interests:** The authors have declared that no competing interests exist.

the presence of metastasis, age greater than 18 months at diagnosis, *MYCN* oncogene amplification, histology, and/or chromosomal ploidy [3]. The current standard therapy for high-risk neuroblastoma includes multi-agent chemotherapy induction and surgical resection of the primary tumor, consolidation chemotherapy, autologous stem cell rescue, and radiation therapy followed by treatment with anti-ganglioside-2 (GD2) immunotherapy, cytokines, and cis-retinoic acid [4]. High-risk group of neuroblastoma accounts for approximately 15% of cancer-related mortality in children with a five-year survival rate of less than 40% [1]. Therefore, there is an unmet need for a more effective novel treatment strategy against this deadly pediatric cancer.

Immunotherapy has proven successful in treating both adult and pediatric cancers [5–11]. Many pediatric solid tumors, including neuroblastoma, have low tumor mutation burdens and decreased or absent surface human leukocyte antigen (HLA) expression that has traditionally rendered them immunologically "cold" [12]. In addition, intensive chemotherapy can further compromise innate and adaptive immune responses [12]. Recently, ch14.18, a monoclonal antibody (mAb) targeting GD2, a ganglioside expressed on neurons, melanocytes, and peripheral nerve fibers, has demonstrated success in clinical treatment by prolonging survival [13–15]. Recent clinical trials have demonstrated improved outcomes with ch14.18 treatment, in combination with interleukin (IL)-2, with or without granulocyte macrophage colony stimulating factor (GM-CSF), and isotretinoin [13, 16]. Furthermore, inhibition of programmed cell death protein 1 (PD-1) has been found to augment the ch14.18-induced immune response against neuroblastoma [17]. Moreover, combination treatment with the anti-GD2 mAb, *dinutuximab*, has been found to prolong survival in *in vivo* mouse models [18]. Monoclonal antibody-based immunotherapy is an important strategy for targeted cancer therapy and recent studies suggest that the identification of additional, novel therapeutic targets may improve treatment outcomes for children with neuroblastoma [5, 9, 19].

Gastrin-releasing peptide receptor (GRP-R) is a G protein-coupled receptor (GPCR) that is overexpressed on the membranes of dysplastic cells in numerous cancers, including neuroblastoma [20, 21]. GPCRs, such as GRP-R, possess seven transmembrane domains and four extracellular domains which facilitate key roles in numerous biologic processes including inflammation and cancer. We have previously demonstrated that GRP-R is upregulated in poorly-differentiated neuroblastoma, promoting oncogenesis via activation of the PI3K/AKT signaling pathway, and plays an important role in the stabilization of the oncoprotein N-myc [22, 23]. N-myc has been shown to contribute to the tumor immunosuppressive environment and cancer immune escape in neuroblastoma [24]. Overexpression of GRP-R has been shown to enhance the stability of the oncoprotein N-myc, whereas knockdown of GRP-R destabilized N-myc in *MYCN*-amplified neuroblastoma cells [25]. Silencing of GRP-R also decreased the activation of AKT2 in the PI3K/AKT pathway, thereby inhibiting mTOR and contributing to the induction of autophagy and decreased tumor growth [26]. Interference of GRP/GRP-R-induced signaling has also been shown to inhibit vascular endothelial growth factor (VEGF)-dependent tumor angiogenesis [27]. In addition to these findings, the location of GRP-R on the tumor cell membrane suggests GRP-R would be an ideal and accessible target for a novel anti-cancer therapy. Furthermore, its extracellular domains (one N-terminus and three extracellular loops) could serve as epitope targets for antibodies, making GRP-R an ideal immune recognition molecule. Hence, we hypothesized that GRP-R could serve as a novel immunotherapeutic target and we sought to evaluate whether GRP-R mAbs could efficaciously inhibit tumorigenesis and metastasis, as well as eliminate residual tumors in high-risk neuroblastoma using *in vitro* and *in vivo* neuroblastoma tumor models.

## Materials and methods

### Antibodies and reagents

Humanized antitumor GRP-R mAb production and initial evaluation were completed by Dr. Ningyan Zhang's group at the University of Texas Health Science Center at Houston. Isotype control antibody (N1-25) is an in-house evaluated monoclonal antibody [28]. GRP-R polyclonal antibody was purchased from Abcam (ab39963, Cambridge, MA). The four peptides of the GRP-R extracellular domains were synthesized with a biotin-tag at the N-terminus by GenScript Biotech (Piscataway, NJ).

Primary antibodies against pAKT (ser473, #9271), AKT (#4685), pH3 (ser10, #3377), Granzyme B (#46890), and pStat1 (Y701, #9171) were purchased from Cell Signaling Technology (Danvers, MA). The primary antibodies of β-actin and GAPDH were from Sigma-Aldrich (St. Louis, MO). Secondary goat anti-human IgG antibody was purchased from Invitrogen (# A18811). Secondary goat anti-mouse and anti-rabbit antibodies were obtained from Santa Cruz Biotechnology, Inc (Santa Cruz, CA). Antibody of PD-L1 (#329702) and IFNγ were purchased from BioLegend (San Diego, CA). Bombesin (BBS) was purchased from Tocris Bioscience (Minneapolis, MN). Human IL-2 was purchased from Pepro Tech (Cranbury, NJ). IFNγ DuoSet ELISA kit was purchased from R&D Systems (Minneapolis, MN). Flu-4 Direct™ Calcium Assay Kit was from Invitrogen (Eugene, OR).

### Cell lines and culture

The human neuroblastoma cell lines BE(2)-C, SK-N-AS, SH-SY5Y, SHEP, SK-N-DZ, and LAN-1 were purchased from the American Type Culture Collection (Manassas, VA). Cells were maintained in Rockwell Park Memorial Institute (RPMI) culture medium 1640 with 10% fetal bovine serum at 37°C in a humidified atmosphere consisting of 5% $CO_2$ and 95% air. Our laboratory previously established the stable luciferase-expressing neuroblastoma cell lines, BE(2)-C/Luc and SK-N-AS/Luc, which were selected and cultured in zeocin (50 μg/ml) contained medium. BE(2)-C/shCON, which functioned as a control and GRP-R shRNA knock-down cells (shGRP-R) were both established previously [26]. Natural killer (NK) cells were purchased from Lonza, cultured in Lymphocyte Growth Media-3 (LGM-3, Lonza), and stimulated by IL-2 (100 U/ml) supplemented in the media. NK cell purity was confirmed with flow cytometry by staining for CD3⁻CD56⁺. These activated NK cells were then used for GRP-R mAb-mediated antibody-dependent cellular cytotoxicity (ADCC) assays.

### Phage library panning and antibody expression

The processes of phage library panning and antibody expression were previously reported [28]. In this study, a mixture of four peptides of GRP-R extracellular domains was used for panning. Briefly, the antigen peptides were coated on a MaxiSorp immune tube overnight and blocked with 5% milk. Then the pre-blocked phage library was incubated with the coated antigen for two hours, and the phage binders were eluted with triethylamine. The antigen-bound phages were screened using enzyme-linked immunosorbent assay (ELISA) and detected using HRP-conjugated mouse-anti-M13 antibody (Santa Cruz, sc-53004). The positive phage clones were constructed into full IgG1 heavy and light chain backbones in a mammalian expression vector for expression in HEK293 cells as we previously reported [28]. IgG antibodies were purified from HEK293 expression culture supernatants using Protein A affinity resin.

## Enzyme-linked immunosorbent assay

A high binding Corning 96-well plate was coated with 5 μg/mL of streptavidin (SA) overnight at 4°C. After SA was aspirated, 100 μL of GRP-R-E1 (5 μg/mL) was added and incubated at room temperature (RT) for one hour. The wells were washed with PBST three times and 100 μL of three-fold serial diluted antibodies were added and the wash was repeated after one hour. Then the anti-human IgG Fab2 HRP-conjugated antibody was diluted 1:5000 and added for incubation at RT for one hour. Following the washing step, TMB substrate was added and incubated for approximately 8 minutes at RT. The reaction was stopped with 50 μL of 1 M $H_2SO_4$ and the OD450 nm was read using a spectrophotometer (Molecular Devices). $EC_{50}$ values were estimated using a three-parameter nonlinear fitting model using GraphPad Prizm, v8.1.

## Affinity measurement with biolayer interferometry

The affinity of antibodies was measured by 8-channel Octet RED96 system (ForteBio). The SA sensors were saturated with 100 nM of GRP-R antigen and followed by incubation in kinetics buffer. Then the loaded sensors were exposed to antibodies in a series of three-fold titrations. The kinetic buffer without antibody was used as correction. The *KD* values were obtained by fitting the data to a 1:1 binding model.

## Cytotoxicity assay and IFNγ releasing measurement

Luciferase-expressing BE(2)-C/Luc and SK-N-AS/Luc cells were plated onto a 96-well white plate (Falcon #353296) at 3,000 and 6,000 cells/well, respectively, for culturing overnight. Then, GRP-R mAb-1 at 2 μg/ml alone, NK cells alone, or GRP-R mAb-1 at 2 μg/ml combined with NK cells, were added to the cultured cells. The ratio of effector cells (NK cells) to target cells (BE(2)-C/Luc) was 2:1. The cells were incubated at 37°C in a humidified atmosphere of 5% $CO_2$ for 4 hours. Luciferin was added to the cell culture medium at a final concentration of 150 ng/ml for 5 minutes. The luminescent signal was then read using the Cytation 5 Cell Imaging Multi-Mode reader (BioTek, Winooski, VT). The cultured medium in the above setup was collected for detection of IFNγ secretion via NK cells using the DuoSet ELISA Development System (IFNγ: DY285, R&D Systems, Minneapolis, MN) according to the manufacturer's protocol.

## Flow cytometry

GRP-R mAb binding affinity was evaluated by flow cytometry. BE(2)-C/shCON and shGRP-R cells were detached using cell dissociation buffer (Gibco #13151014). After washing them with 1% bovine serum albumin (BSA) in PBS, cells were incubated with primary antibody for 60 minutes at 4°C. After washing with 1% BSA in PBS, cells were incubated with Alexa Fluor 488 –tagged secondary antibody for 30 minutes at 4°C. Human IgG1 non-binder antibody was used for isotype binding control. PD-L1 membrane expression was evaluated by staining cells with primary PD-L1 antibody (Biolegends, #329702), followed by anti-mouse Alexa Fluor 594 conjugated secondary antibody staining. Stained cells were analyzed with the flow cytometer (BD Accuri C6+, BD Biosciences, Franklin Lakes, NJ).

## Cell growth and anchorage-independent growth assay

Neuroblastoma cells were plated onto 96-well plates at 3,000–6,000 cells/well for overnight. GRP-R mAbs-1 was added to the culture media at various concentrations for 72 hours. The cell growth rate was measured using the Cell Counting Kit-8 (Dojindo Molecular Technologies,

Inc, Rockville, MD). For the anchorage-independent growth assay, cells were trypsinized and resuspended in RPMI medium 1640 containing 0.4% agarose and 5% FBS. BE(2)-C cells were overlaid onto a bottom layer of solidified 0.8% agarose in RPMI medium 1640 containing 5% FBS and incubated for 2 weeks. Colonies were formed in the soft agar and stained with 0.05% crystal violet, photographed, and quantified.

## Immunoblotting

Cells were collected using cell lysis buffer and denatured protein samples were prepared for immunoblotting. Equal amounts of protein were loaded and separated by NuPAGE 4–12% Bis-Tris gel, followed by transfer onto PVDF membranes (Bio-Rad, Hercules, CA, USA). Membranes were blocked with 5% nonfat milk in TBS-T for 1 hour at room temperature. The blots were then incubated with antibodies against the human target proteins by using rabbit or mouse anti-human antibodies (1:500–2000 dilution) overnight at 4°C. Anti-rabbit or anti-mouse secondary antibodies conjugated with HRP were incubated for 1 hour and visualized using an enhanced chemiluminescence detection system (PerkinElmer, Waltham, MA, USA).

## Immunohistochemistry and immunofluorescence staining

Immunohistochemical (IHC) staining was performed using DAKO EnVision+ System-HRP from Dako North America, Inc. (Carpinteria, CA). Mouse neuroblastoma xenografts were excised and fixed in 10% buffered formalin overnight and embedded in paraffin wax. Tumor sections (5 μm) were mounted on glass slides. Samples were deparaffinized and rehydrated. The antigen was retrieved using 0.01 M sodium-citrate buffer (pH 6.0) at a sub-boiling temperature for 10 minutes after boiling in a microwave oven. To block endogenous peroxidase activity, the sections were incubated with 3% hydrogen peroxide for 5 minutes. After 1 hour of pre-incubation in 5% normal goat serum to prevent nonspecific staining, the samples were incubated with the primary antibody against GRP-R or Granzyme B at 4°C overnight. They were then washed with buffer three times for 5 minutes each and incubated with secondary antibody for 30 minutes at RT. Sections were developed with the DAB reagent. The reaction was terminated by immersing slides in $dH_2O$, and sections were counterstained with hematoxylin. Slides were then dehydrated with ethanol and xylene. Coverslips were mounted and slides were left to dry. The IHC images were taken using a Cytation 5 Cell Imaging Multi-Mode reader. For tumor cell mitosis detection using immunofluorescence, paraffin-embedded sections were stained with anti-human phospho-Histone H3 (Ser10) antibody followed by Alexa Fluor 568 Dye (Life Technologies, Grand Island, NY). DAPI was used for staining nuclei. Images were captured using a Cytation 5 Cell Imaging Multi-Mode reader.

## RNA isolation and quantitative polymerase chain reaction with RT-PCR

Total RNA was isolated and purified using a Trizol Reagent (Thermo Scientific). The High-Capacity cDNA Reverse Transcription Kit (Applied Biosystems, Carlsbad, CA) was used to synthesize complementary DNA. Reverse transcription polymerase chain reaction (RT-PCR) was performed using iTaq Universal SYBR Green Supermix (Bio-Rad, Hercules, CA), and data were collected with a CFX96 instrument (Bio-Rad). Results were normalized to an endogenous control, glyceraldehyde 3-phosphate dehydrogenase (GAPDH). Amplification was performed for 40 cycles of 30 seconds at 95°C, 30 seconds at 55°C, and 40 seconds at 72°C. Primers used to detect the expression of qPCR were the following: PD-L1-fw: `TGGTGTAG CACTGACATTCA`; PD-L1-rv: `TCCAATGCTGGATTACGTCT`; and GAPDH-fw: `CCACTCCTC CACCTTTGAC`; GAPDH-rv: `ACCCTGTTGCTGTAGCCA`.

### Cytotoxicity assay in murine xenograft model

All studies were approved by the Institutional Animal Care and Use Committee (IACUC) at UT Southwestern Medical Center. All animals were maintained in a pathogen-free environment at Animal Resource Center facility of UT Southwestern Medical Center. BE(2)-C/Luc human neuroblastoma cells at $1 \times 10^6$ cells/100 ul of HBSS medium were injected subcutaneously into the flanks of 4–6 week old male athymic nude mice for tumor formation. When subcutaneous tumors formed and were visible at a diameter of approximately 7 mm, the mice were randomized to four treatment groups: IgG (4 mg/kg), GRP-R mAb (4 mg/kg), IgG plus NK cells (0.1 million), or GRP-R mAb plus NK cells (0.1 million). Mice were treated twice per week for three weeks. Mice were monitored daily for xenograft formation, stress or suffering. Tumor growth was assessed by measuring the two greatest perpendicular tumor diameters with vernier digital calipers (Neiko Tools USA). Xenograft volumes were estimated using the following formula: $[(\text{length} \times \text{width}^2)/2]$. Weight and tumor volume were recorded biweekly. Mice were euthanized four weeks after injection when they met institutional euthanasia criteria for tumor volume greater than 2000 mm$^3$, 20% loss of body weight, and ulceration or severe necrosis of tumor. Mice were euthanized with $CO_2$ asphyxiation followed by cervical dislocation. The tumors were excised, weighed, and fixed in 10% buffered formalin. Tumor tissues were further processed for embedding in paraffin, sectioned, and stained with hematoxylin and eosin at the Molecular Pathology Core Histology Service Center at UT Southwestern Medical Center.

### Statistical analysis and experimental analysis

All experiments were repeated in triplicate. The scoring index and relative expression values were expressed as mean ± SEM; statistical analyses were performed using Student's t-test for comparisons between the groups. A $p$-value of $< 0.05$ was considered significant.

## Results

### GRP-R mAb production

A GRP-R mAb screen was performed as summarized in Fig 1A. Four peptides of the GRP-R extracellular domains (E1, E2, E3, and E4) with a biotin-tag at the N-terminus were synthesized and used as antigens to screen the phage library for scFv binders (Fig 1B). We initially obtained four mAbs against the extracellular domain 1 (E1) of GRP-R. These four scFvs (mAb-1, mAb-17, mAb-21, and mAb-44) were cloned into a mammalian expression vector for antibody production as previously reported [29]. The purified antibodies were electrophorized on SDS-PAGE at non-reduced (-DTT) conditions, heavy chain (55 kDa) and light chain (25 kDa) were separated at reduced conditions (+DTT) (Fig 1C). The *KD* values were obtained by fitting the data to a 1:1 binding model (S1 Fig). The EC$_{50}$ of each mAb was determined with titration ELISA (Fig 1D), and the binding specificity was confirmed (S2 Fig). Our results showed that GRP-R mAb-1 has the highest efficacy with the lowest EC$_{50}$ at 4.607 ng/ml and was a potential candidate for further functional studies.

### Inhibitory effects of GRP-R mAb on human neuroblastoma cells

Increased GRP-R expression is associated with the aggressive phenotype of neuroblastoma [20]. Immunoblotting was used to evaluate for GRP-R expression in six human neuroblastoma cell lines. We found that GRP-R was expressed in the *MYCN*-amplified cell lines BE(2)-C, LAN-1 and SK-N-DZ, and the non *MYCN*-amplified cell lines, SHEP, SHSY5Y and SK-N-AS (Fig 2A). We examined the binding specificity of GRP-R antibodies by flow cytometry in BE

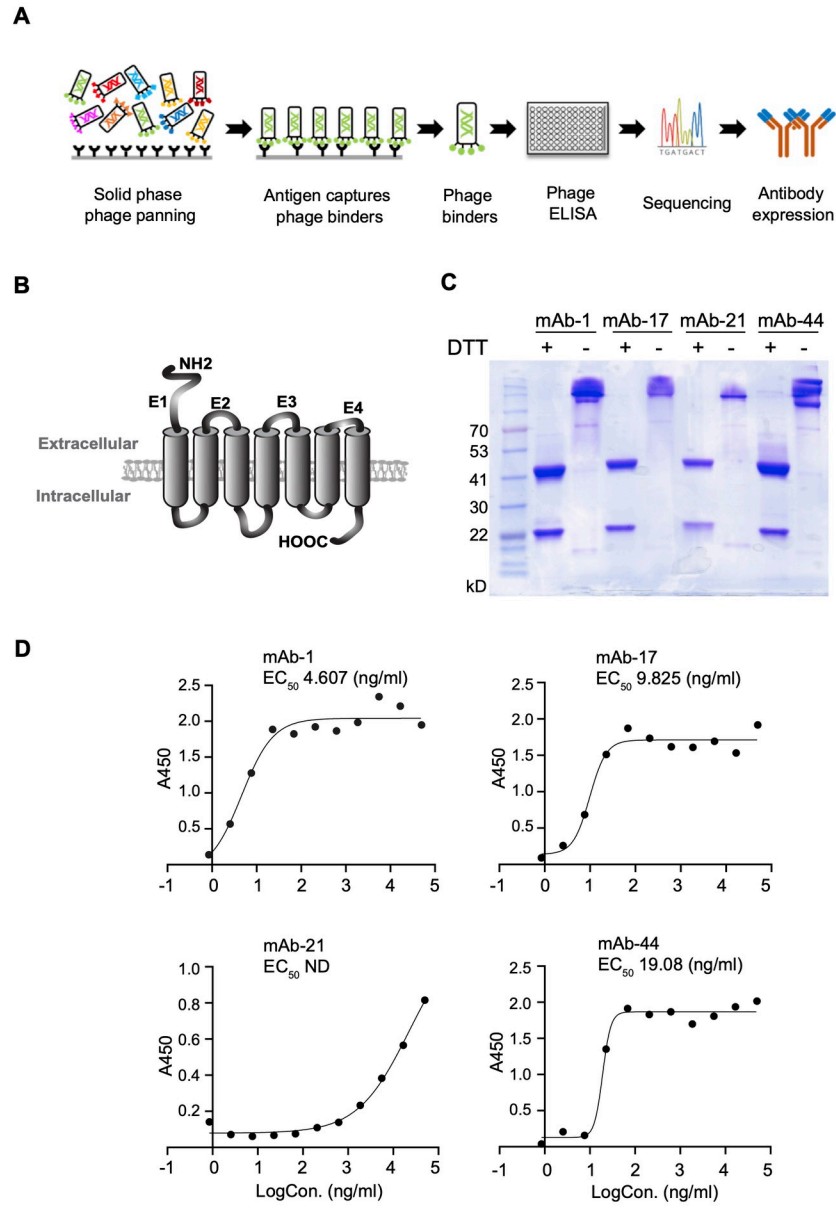

**Fig 1. GRP-R monoclonal antibody development and production.** (A) Human scFv phage display library screening strategy. (B) GRP-R is a G-protein coupled receptor, with seven transmembrane domains, four intracellular and four extracellular domains. We synthesized four peptides of extracellular domains (E1 to E4) for the screening phage library. (C) Purified GRP-R monoclonal antibodies were electrophorized on SDS-PAGE at non-reduced (-DTT) conditions, and heavy chain (55 kDa) and light chain (25 kDa) were separated at reduced condition (+DTT). (D) $EC_{50}$ of each GRP-R mAb was determined with titration ELISA.

(2)-C cells (shCON), as well as in GRP-R silenced BE(2)-C cells (shGRP-R) (Fig 2B). Our results showed that mAb-1, mAb-21 and mAb-44 demonstrated histogram shift in mean fluorescence intensity (MFI) between shCON and shGRP-R cells (Fig 2C), and that mAb-1 has the highest value of fluorescence intensity at 644,000 (Fig 2D). In addition, we obtained four human neuroblastoma patient-derived xenografts (PDXs) from the Children's Oncology Group (COG) derived from patients with stage IV, high-risk disease, and stained them with

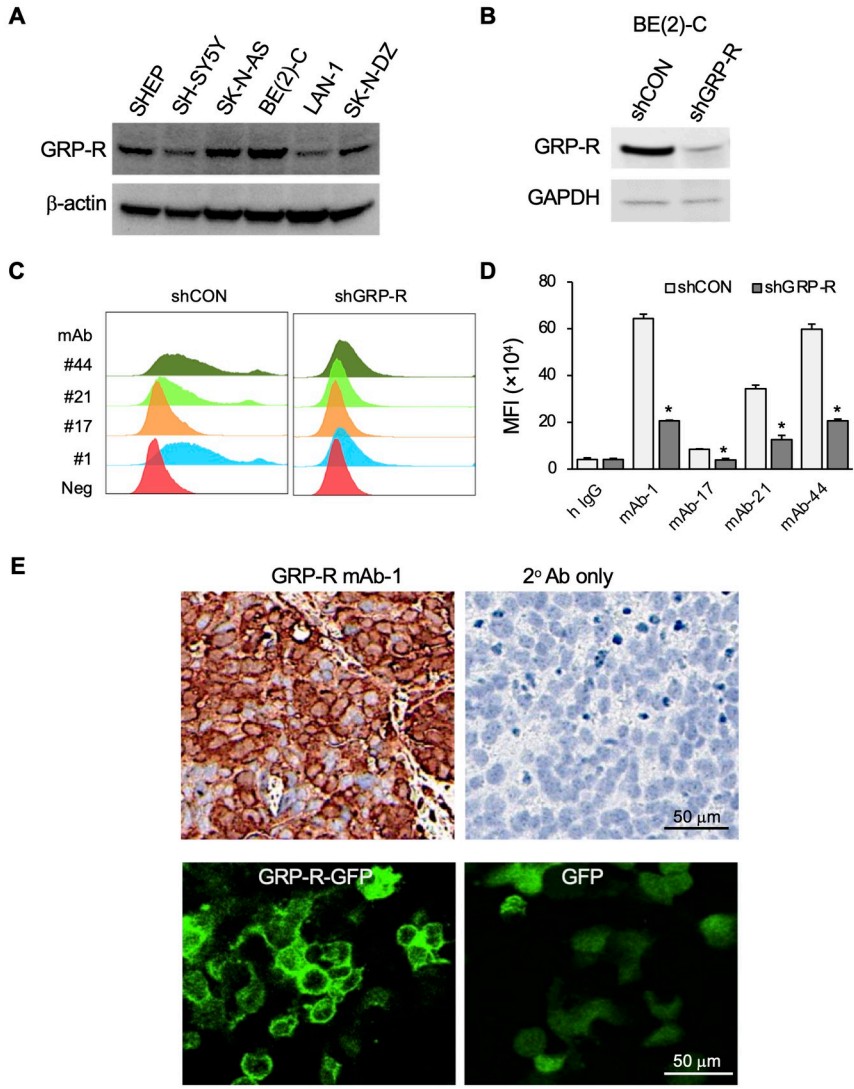

**Fig 2. Evaluation of GRP-R mAb binding affinity with neuroblastoma cells and tissue.** (A) GRP-R expression was detected by immunoblotting in six human neuroblastoma cell lines. β-actin was used for protein loading control. (B) GRP-R knock down with shRNA was confirmed with immunoblotting. GAPDH was used for protein loading control. (C) The antibody binding assay was carried out with control cells BE(2)-C/shCON and GRP-R knockdown cells BE (2)-C/shGRP-R by flow cytometry. Total cell number $5 \times 10^5$ was analyzed. Primary antibody concentration was 25 μg/ml. Secondary antibody (tagged with Alexa Fluor 488) concentration was 5 μg/ml. (D) The mean fluorescence intensity (MFI) in BE(2)-C/shCON and shGRP-R cells was quantified. (E) Immunohistochemical staining on neuroblastoma PDX tissue with GRP-R mAb-1, showed the specific staining of GRP-R on cytoplasmic membrane (dark brown on the cytoplasmic membrane, top left), and the absence of staining of negative control with secondary antibody only (top right). Exogenously expressed GRP-R-GFP was localized on cytoplasmic membrane (bottom left) and GFP alone was non-specifically distributed in cytoplasm.

GRP-R mAbs. Tissue samples stained with the secondary antibody alone were used as a negative control. Our results revealed that GRP-R mAb-1 binds to neuroblastoma PDX tissue with high specificity, as demonstrated by the positive staining of the cytoplasm membrane, with the absence of brown staining on tissues treated with the secondary antibody alone (Fig 2E). Thus, the GRP-R mAb-1 was selected for further *in vitro* and *in vivo* functional studies.

## GRP-R mAb-1 blocks bombesin-induced PI3K/AKT activation and inhibits cell proliferation and anchorage-independent growth

GRP, the ligand of GRP-R, functions as a mitogen to induce cell cycle progression through activation of the PI3K/AKT pathway. To examine the GRP-R mAb-1 inhibitory function on GRP/GRP-R downstream signaling and neuroblastoma tumor cell growth, we pre-treated BE(2)-C/shCON and shGRP-R cells with GRP-R mAb-1 (1 μg/ml) one hour prior to bombesin (a GRP analog) treatment. As shown in Fig 3A, GRP-R mAb-1 inhibited bombesin-induced activation of PI3K/AKT, as indicated by a decreased phosphorylated AKT level. The same result was obtained using SK-N-AS cells (Fig 3B). Furthermore, GRP-R mAb-1 reduced the BBS stimulated downstream intracellular $Ca^{2+}$ releasing signals (Fig 3C).

In order to assess the effects of GRP-R mAb-1 on cellular proliferation BE(2)-C cells were treated with GRP-R mAb-1 at increasing doses and cultured for 4 days. Cellular proliferation decreased by 30–43% compared to control cells in a dose-dependent manner (Fig 3D). As shown in Fig 3E, an anchorage-independent colony growth assay demonstrated that co-incubation of neuroblastoma cells with GRP-R mAb-1 significantly decreased the colony number by 39% in control cells (pCMV) and 50% in GRP-R overexpression cells (pCMV-GRPR). Taken together, these findings demonstrate that GRP-R mAb-1 has an antagonist role by blocking bombesin-stimulated GRP-R activation, in addition to inhibiting cellular proliferation and anchorage-independent growth.

## GRP-R mAb mediates ADCC on GRP-R expressing neuroblastoma cells and induces IFNγ secretion from NK cells

Antibody therapy has therapeutic advantages over small molecule inhibitors. Antibody-dependent cellular cytotoxicity (ADCC) is one of the most important mechanisms behind the anti-tumor effect of mAbs. To evaluate the specific killing of tumor cells, a modified bio-luminescence-based cytotoxicity assay was used instead of a standard chromium release assay. We previously established the stable luciferase-expressing neuroblastoma cell lines, BE(2)-C/Luc and SK-N-AS/Luc, which have been used to monitor tumor growth and metastasis using *in vivo* models [30, 31]. We treated BE(2)-C/Luc and SK-N-AS/Luc cells with GRP-R mAb-1 at 2 μg/ml alone, NK cells alone, or GRP-R mAb-1 at 2 μg/ml combined with NK cells. Our results showed that adding NK cells alone (at a ratio of 2:1 effector NK cells to target BE(2)-C/Luc) caused cell death to 58.96% of BE(2)-C/Luc cells. The addition of GRP-R mAb-1 (2 μg/ml) significantly increased cell death to 73.19% within 4 hours (Fig 4A, **left panel**). Representative cell images of the ADCC assay are shown in Fig 4A, **right panel**. In addition, we obtained similar results using the cell line SK-N-AS/Luc in an ADCC assay (Fig 4B). Our results indicate that ADCC contributes to the antitumor action of GRP-R mAb through NK cells. Interestingly, we found that the number of surviving tumor cells was slightly increased after 24 hours, compared to 4 hours post co-culture with GRP-R mAb-1 and NK cells, and the efficiency of ADCC was increased at a higher ratio (5:1) of effector NK cells to target BE(2)-C/Luc (Fig 4C). Our results indicate that the action of ADCC was inhibited after some time.

Cytokine IFNγ is one of the key factors secreted by immune cells, such as NK cells and T cells, in the tumor microenvironment [32]. Therefore, we examined the effects of IFNγ secretion in the supernatant of co-cultured BE(2)-C and NK cells, with or without GRP-R mAb-1 (2 μg/ml). Our results showed that IFNγ was detected in the supernatant of the co-culture after 24 hours, and GRP-R mAb-1 increased the secretion of IFNγ approximately 1.6-fold from 4.4 pg/ml to 7.2 pg/ml (Fig 4D).

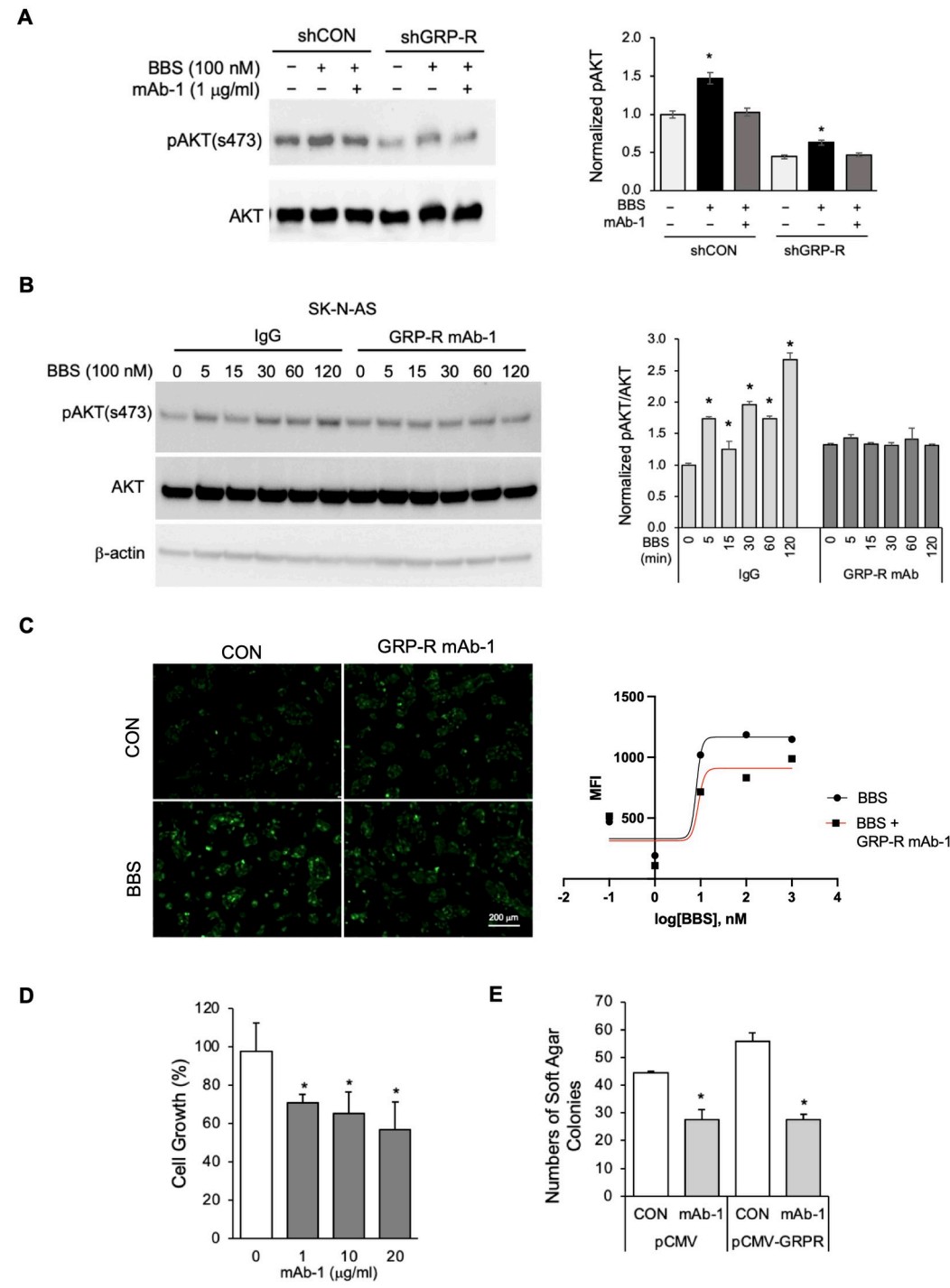

**Fig 3. GRP-R mAb-1 inhibitory roles on neuroblastoma cell *in vitro*.** (A) GRP-R mAb-1 inhibited BBS-mediated activation of AKT in BE(2)-C cells. The phosphorylation of AKT (S473) was detected by immunoblotting assay in BE(2)-C cells. The cells were pretreated with GRP-R mAb-1 at 1 mg/ml for 60 minutes, then treated with BBS at 100 nM for 15 minutes (left). The level of pAKT (s473) was normalized to total AKT with ImageJ (right). (B) GRP-R mAb-1 inhibited BBS-mediated activation of AKT in SK-N-AS cells. The phosphorylation of AKT (S473) was detected by immunoblotting assay in SK-N-AS cells. The cells were pretreated with GRP-R mAb or its isotype control IgG for 60 minutes, followed by BBS at 100 nM for 5–120 minutes (left). The level of pAKT (s473) was normalized to total AKT with ImageJ (right). (C) GRP-R mAb-1 reduced BBS-stimulated intracellular $Ca^{2+}$ release signals. SK-N-AS cells were treated with BBS at 100 nM for 30 minutes. The calcium response was subsequently analyzed with Flu-4 Direct™ Calcium Assay Kit. The green fluorescence signal indicated the released intracellular $Ca^{2+}$ in SK-N-AS cells (left). The mean of green fluorescence

intensity was quantified with Gen5 Software (right). (D) GRP-R mAb-1 inhibited neuroblastoma tumor cell growth. BE(2)-C cells were treated with GRP-R mAb-1 and cultured for 4 days. Cell growth was measured with CCK-8 kit. (E) GRP-R mAb-1 decreased the capability of anchorage-independent in BE(2)-C cells. The cells were cultured in soft agar with or without GRP-R mAb-1 (1 μg/ml) for 2 weeks. Data represent mean ± STDEV; * = *p*<0.05 vs. CON.

## Immune checkpoint protein PD-L1 expression in neuroblastoma cells

In the tumor microenvironment, IFNγ can induce tumor cells to express PD-L1 and PD-L2, which subsequently interact with their receptor PD-1 on immune cells, inducing apoptosis of PD-1 positive immune cells, leading to immune suppression and evasion [33]. We treated BE

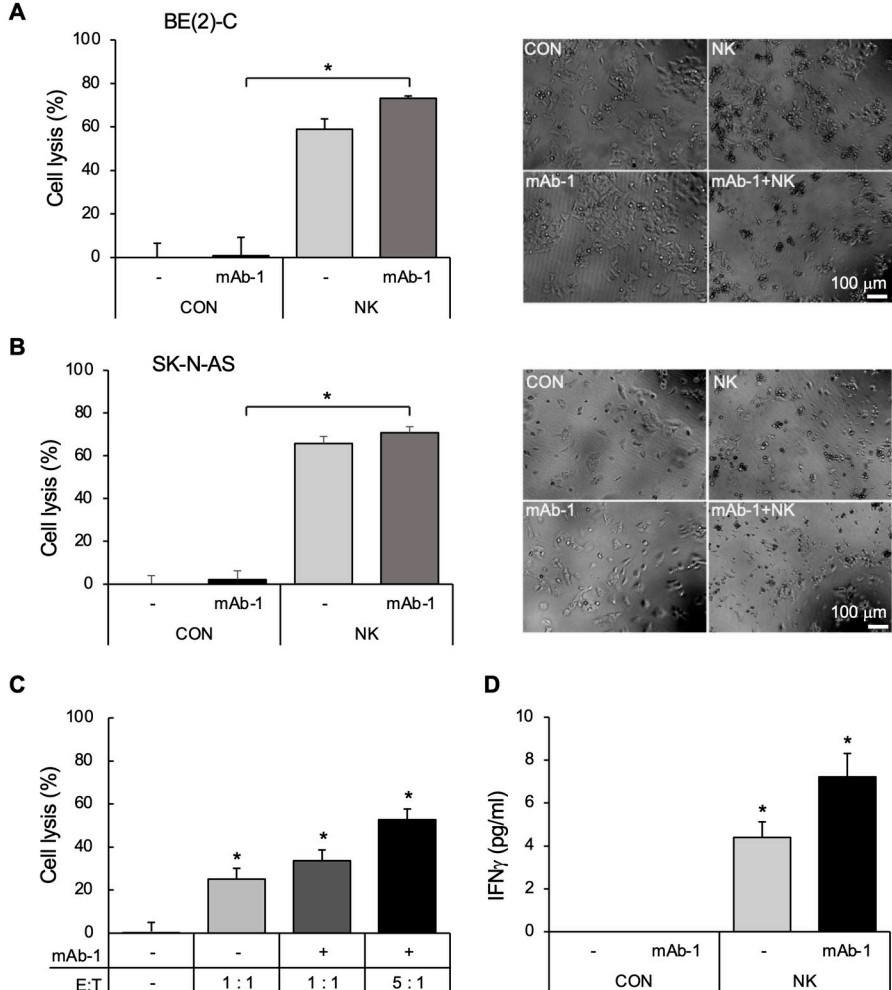

**Fig 4. GRP-R mAb–mediated antibody-dependent cellular cytotoxicity (ADCC) in neuroblastoma.** (A) BE(2)-C/ Luc as target cells were co-cultured with effector NK cells (E:T at 2:1) with or without GRP-R mAb-1 at 2 μg/ml for 4 hours. The percent of lysed cells was calculated by measuring the luciferase activity of survived cells in control samples and treated samples (Left). The typical cell images were taken under microscope, showing the integral and killed cells (right). (B) ADCC assay in SK-N-AS cells. SK-N-AS/Luc as target cells were co-cultured with effector NK cells (E:T at 2:1) with or without GRP-R mAb-1 at 2 μg/ml for 4 hours (Left). The typical cell images were taken using microscopy, showing the integral and killed cells (right). (C) ADCC assay in BE(2)-C cells. The cells were co-cultured with effector NK cells with or without GRP-R mAb-1 at 2 μg/ml for 24 hours. (D) The secretion of IFNγ in the co-culture media was measured with ELISA kit. BE(2)-C cells were co-cultured with NK cells (E:T at 2:1) with or without GRP-R mAb-1 (2 μg/ml) for 24 hours. Data represent mean ± STDEV; * = *p*<0.05 vs. CON.

(2)-C and SK-N-AS cells with IFNγ at 10 ng/ml for 24 hours and found that IFNγ induced the expression of PD-L1 in both cell lines showed with histogram shift by flow cytometry analysis and fluorescent microscopy images (Fig 5A). IFNγ increased the transcriptional level of both PD-L1 and PD-L2 in BE(2)-C cells, PD-L1 expression was increased 14-fold, whereas PD-L2 expression was only increased 1.6-fold, as measured by qRT-PCR (S3 Fig).

We then examined the expression of PD-L1 in six neuroblastoma cell lines and found that the expression of PD-L1 was induced by IFNγ in a dose-dependent response in the levels of both protein (Fig 5B) and mRNA (Fig 5C), in all except SH-SY5Y cell line. The expression of PD-L2, however, demonstrated no change or a weak increase in expression in response to IFNγ treatment (S4 Fig), consistent with previous studies [34].

The activation of Stat1 upregulates the expression of PD-L1 and PD-L2 in gastric cancer [35]. In order to confirm that IFNγ induced PD-L1 via Stat1, we treated six neuroblastoma cell lines with IFNγ at two doses (1 and 10 ng/ml) for 24 hours. Our results showed that the phosphorylation of Stat1 increased in a dose-dependent fashion in response to IFNγ treatment in the BE(2)-C, SK-N-AS, SHEP, SK-N-DZ and LAN-1 cell lines (Fig 5D). However, the SH-SY5Y cells did not demonstrate activation of Stat1 in response to IFNγ. Our results suggest that IFNγ induces the expression of PD-L1 via activation of Stat1 in neuroblastoma.

## GRP-R mAb-1 induced ADCC in mouse xenografts tumor tissue

To evaluate the antitumor function of GRP-R mAb *in vivo*, we examined the effects of treatment with GRP-R mAb, NK cells, or combination treatment with GRP-R mAb-1 and NK cells. First, we used the BE(2)-C cell line to establish xenografts in athymic nude mice as described previously [26]. Ten days after tumor cell injection, when subcutaneous tumors formed and were visible at a diameter of approximately 7 mm, GRP-R mAb-1 and NK cells were subsequently administrated into the subcutaneous tumors. The effect of GRP-R mAb-1 mediated ADCC was examined by performing immunohistochemical staining of granzyme B in the tumor tissue sections using anti-Granzyme B antibody. As expected, granzyme B was detected surrounding the small immune NK cells in the tumors treated with GRP-R mAb + NK, but not in the other three treatment groups (CON, mAb, and IgG+NK) (Fig 6A). The detection of cytotoxic granzyme B in only the tumor tissues treated with both GRP-R mAb and NK cells indicated that GRP-R mAb induced ADCC in tumors, consistent with our *in vitro* results. Furthermore, we stained tumor tissue sections with the anti-human phospho-Histone H3 (ser10) antibody followed by Alexa Fluor 568 Dye and DAPI (49,6- diamidino-2-phenylindole dihydrochloride) to stain the nuclei. Phospho-Histone H3 (Ser10) is a cell mitosis marker tightly correlated with chromosome condensation during both mitosis and meiosis [36]. As shown in Fig 6B, there were only 40% and 52.4% mitotic cells in the tumor sections treated with GRP-R mAb + NK cells and IgG + NK cells, respectively when compared to control. There was also no difference in tumors treated with GRP-R mAb alone relative to control, indicating the importance of effector cells in the tumor microenvironment on mAb-induced ADCC. Interestingly, we found that PD-L1 expression in tumor tissue sections could only be detected in cells surrounded by NK cells (Fig 6C), further suggesting that the expression of PD-L1 is dependent on the tumor microenvironment and IFNγ secreted by NK cells.

## Discussion

Neuroblastoma has traditionally been considered an immunologically "cold" tumor, however, the recent success of mAb ch14.18 in clinical treatment has provided invaluable information regarding augmentation of the immune response and limiting immune escape in neuroblastoma [16]. Despite these recent research advances, high-risk neuroblastoma remains difficult

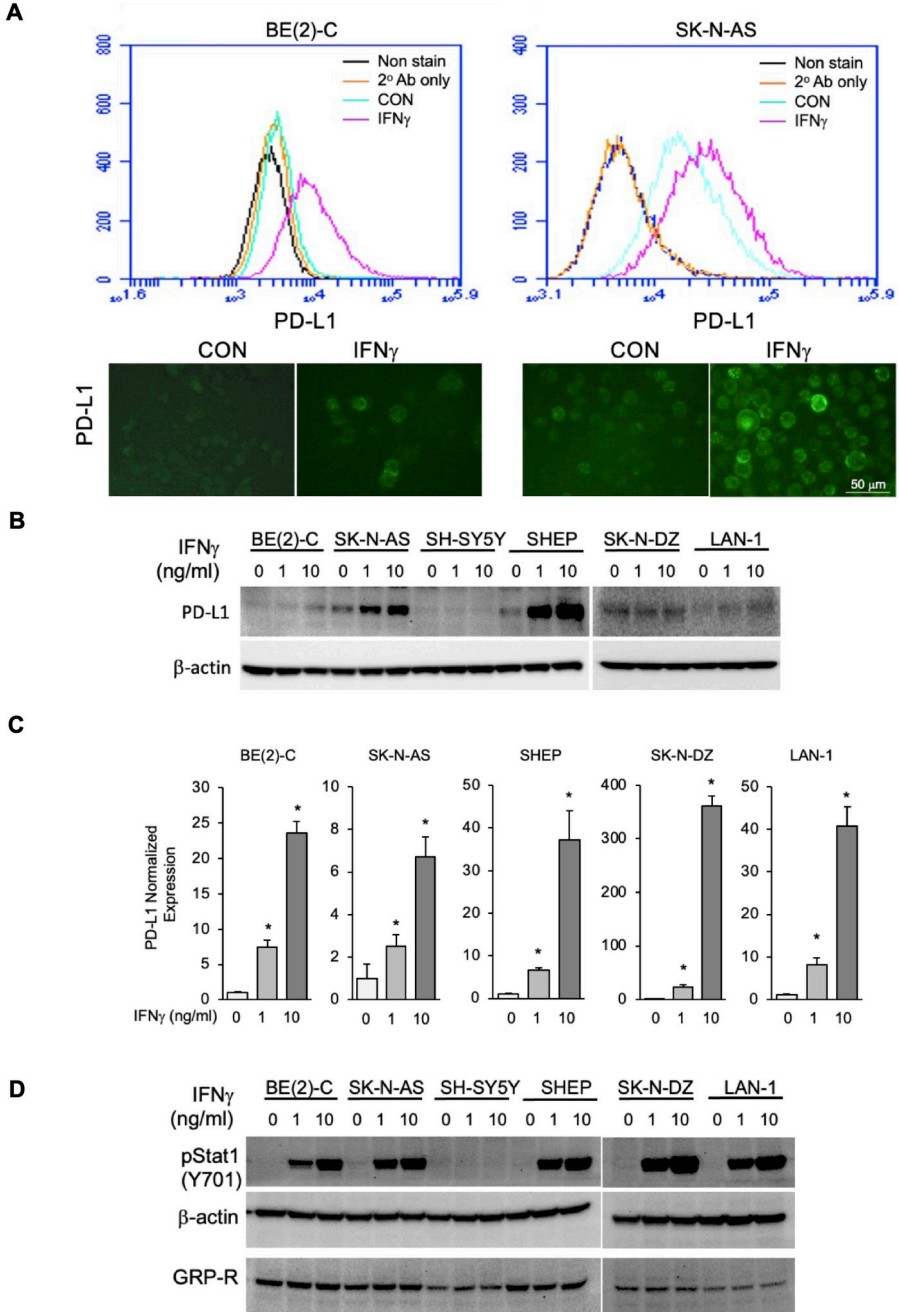

**Fig 5. Immune checkpoint protein PD-L1 expression in neuroblastoma cells.** (A) PD-L1 expression on the surface of BE(2)-C and SK-N-AS cells was analyzed by flow cytometry (top) and fluorescent microscopy (bottom). Cells were treated with IFNγ (10 ng/ml) for 24 hours. (B) The total expression of PD-L1 was detected in six human neuroblastoma cell lines by immunoblotting analysis. The cells were treated with IFNγ at 0, 1, or 10 ng/ml for 24 hours. Total cell lysates were used in immunoblotting assay. β-actin was used for protein loading control. (C) The mRNA level of PD-L1 was measured by qRT-PCR and normalized to GAPDH in six human neuroblastoma cell lines. All cells were treated with IFNγ at 0, 1, or 10 ng/ml for 24 hours. (D) Phosphorylated Stat1 was measured in six neuroblastoma cell lines. All cells were treated with IFNγ at 0, 1, 10 ng/ml for 24 hours. The levels of phospho-Stat1 and GRP-R were detected with immunoblotting. β-actin was used for protein loading control.

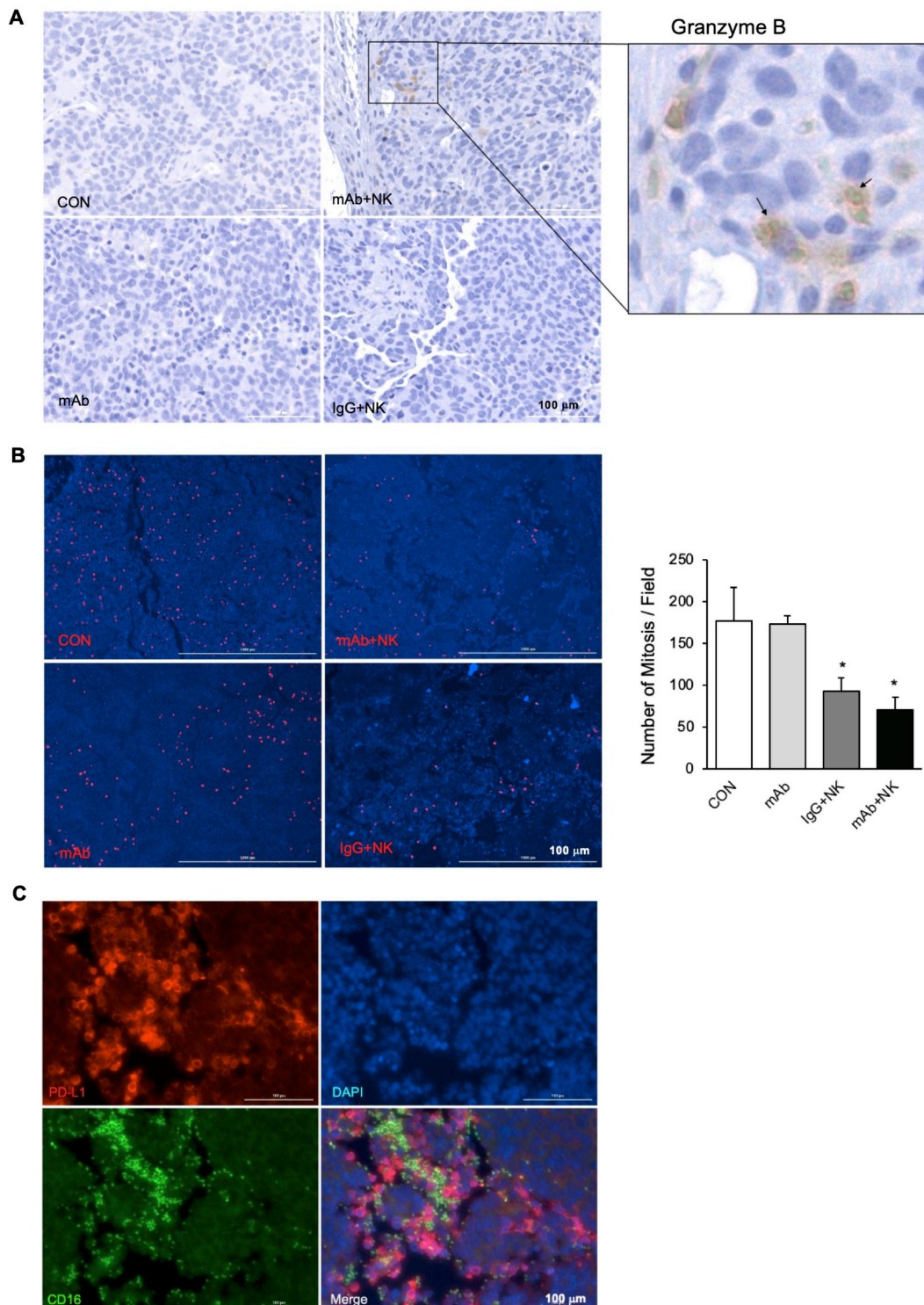

**Fig 6. GRP-R mAb-mediated ADCC in mouse xenograft tumors.** (A) Granzyme B was detected by immunohistochemical staining with Granzyme B antibody in paraffin-embedded sections of BE(2)-C xenografts. (B) Immunofluorescence double staining of Phospho-Histone H3 (red) in the mitotic tumor cells and DAPI on nuclei DNA in all cells (blue) was performed in paraffin-embedded sections of tumor xenografts (left). Mitotic cells were quantified by counting positive immunofluorescent stained phospho-Histone H3 (ser10) cells (right). (C) Immunofluorescence triple staining of PD-L1 on the membrane of tumor cells (red) and CD16-FITC on NK cells (green) in tumor tissue sections. DAPI was used for counterstaining nuclei DNA in all cells. Scale bar: 100 mm. Data represent mean ± STDEV; * = $p<0.05$ vs. CON.

to treat, necessitating the development of novel treatment strategies. Herein, we demonstrate a novel treatment strategy using GRP-R mAb against neuroblastoma.

The overexpression of GRP-R on the cellular membrane of neuroblastoma cells and its role in the modulation of the PI3K/AKT pathway make it an ideal target for novel anti-cancer therapies. GRP-R is a GPCR with seven transmembrane and four extracellular domains. These extracellular domains have the potential to serve as epitopes for antibodies (44). GRP-R-based pharmaceuticals and specific radioactive, cytotoxic, and nonradioactive GRP-analogs have been designed and tested in various animal tumor models with the aim of receptor targeting for tumor diagnosis or anti-cancer therapy [20, 37–41]. However, despite the fact that approximately 30–50% of marketed drugs target a GPCR, mAbs to GPCRs have yet to be approved for clinical use by the FDA [42]. Although small molecule drugs targeting GPCRs have been developed and preclinically tested, antibody therapy has therapeutic advantages over small molecules such as reduced dosing and peripheral restriction [43]. Several mAbs against different GPCRs are currently undergoing testing in phase I and phase II trials involved in cancer and inflammation [42]. Together, these findings support the use of a GRP-R mAb as a potentially novel immunotherapy against neuroblastoma. In the present study, GRP-R mAbs were identified and found to specifically bind to neuroblastoma tissue, inhibit PI3K/AKT activation, and inhibit cell proliferation and anchorage-independent growth. In addition, GRP-R mAb induced mAb-mediated ADCC and, in combination treatment with NK cells, successfully inhibited tumor growth *in vivo*.

In order to create mAb to GRP-R, a phage library was screened for scFv binders and four mAbs against GRP-R extracellular domain 1 (E1) were identified. GRP-R mAb-1 demonstrated the highest affinity at the lowest $EC_{50}$ and was used for further testing. Our results demonstrated GRP-R mAb-1-specific binding to neuroblastoma PDX tissue. Further, we found that treatment with GRP-R mAb-1 inhibits activation of PI3K/AKT, cellular proliferation and anchorage-independent growth. These findings are in line with findings from our previous study which demonstrated decreased tumor growth and inhibition of the PI3K/AKT pathway with GRP-R silencing [26]. Our results demonstrate that anti-GRP-R mAb effectively and significantly inhibits neuroblastoma tumorigenesis in preclinical testing.

The cytotoxic effects of mAb are enhanced by the effector cells, such as NK cells, of the tumor microenvironment [44]. Therefore, we sought to evaluate the effects of GRP-R mAb-1 treatment on the tumor microenvironment. Combination treatment with GRP-R mAb-1 and NK effector cells in a 2:1 ratio significantly induced ADCC in both BE(2)-C and SK-N-AS cell lines. Adding NK cells with anti-GRP-R mAb treatment significantly enhanced cell death when compared to anti-GRP-R mAb treatment alone. However, we found that GRP-R mAb-mediated induction of ADCC was inhibited at 24 hours compared to 4 hours, suggesting the actions of ADCC can be inhibited over time due to NK cell dysfunction [45].

PD-L1 is one of the immune checkpoint proteins that is regulated by IFNγ via JAK/STAT pathway [35]. In order to evaluate the potential role of immune checkpoint pathways in the reduction of ADCC over time, the effects of IFNγ treatment on PD-L1 and PD-L2 expression as well as Stat1 activation in neuroblastoma cells was evaluated. Treatment with IFNγ increased the activation of Stat1, a key upstream regulator of PD-L1, as well as the PD-L1 in a dose-dependent manner. PD-L2 expression did not respond the same as PD-L1, suggesting combination treatment with GRP-R mAb-1 induces ADCC and the secretion of IFNγ. Given findings of the effects of GRP-R mAb-1 + NK cell treatment *in vitro*, the effects of combination treatment on tumor growth *in vivo* were evaluated using an animal tumor model. Treatment with GRP-R mAb-1 in combination with NK cells significantly induced cell death in tumor tissues, however treatment with GRP-R mAb-1 alone did not impair tumor growth.

The positive association between PD-L1 expression and tumor aggressiveness has been demonstrated in numerous cancers including osteosarcoma, breast cancer, ovarian cancer, extrahepatic cholangiocarcinoma, and melanoma [46, 47]. However, these reports can be conflicting. Hamanishi et al suggested that PD-L1 may directly suppress antitumor CD8-positive T cells in ovarian cancer, making the PD-1/PD-L1 pathway an important target for restoring antitumor immunity [46]. Findings in the present study suggest that the varying PD-L1 expression, the effectiveness of current immunotherapies, and the overall clinical outcomes may be tumor microenvironment dependent. Together, these findings suggest that the modulation of the tumor microenvironment and immune checkpoint pathways are vital in preventing immune escape, enhancing cytotoxicity against tumor cells, and limiting tumorigenesis.

Despite the success of GRP-R mAb treatment in combination with NK cells *in vivo* and *in vitro*, further studies are needed to assess the safety and efficacy of GRP-R mAb-1 + NK cells in the clinical setting. There have been several studies evaluating the role of anti-GD2 mAb in combination with NK cells. Tran et al found that concurrent treatment with NK cells increased the efficacy of *dinutuximab* preclinically [48]. Clinical trials evaluating combination anti-GD2 treatment with NK cells have also shown antitumor activity [49, 50]. The success of treatment with anti-GD2 mAb in combination with NK cells in previous studies suggests GRP-R mAb is a potentially novel immunotherapeutic treatment against neuroblastoma that would benefit from clinical testing.

There are several limitations to the present study including the pre-clinical nature of the study. In addition, the significant heterogeneity of neuroblastoma has previously been reported by others [51]. To help mitigate the effects of this heterogeneity, both *MYCN* and non-*MYCN*-amplifying cell lines were used throughout the study in both *in vitro* and *in vivo* experiments to allow for a broader application of our findings. Future directions, however, include clinical trials to establish the safety and effectiveness of GRP-R mAbs for the treatment of neuroblastoma. In this study, four mAbs were obtained against E1 of GRP-R. Future studies would also allow for the production and evaluation of mAbs against GRP-R's three remaining extracellular domains. Our results contribute to previous studies demonstrating the significant impact of the tumor microenvironment on prognosis, the effectiveness of therapies, and clinical outcomes. Further research should be directed to expanding our knowledge of the neuroblastoma tumor microenvironment and ways to effectively modify it to enhance the treatment effect.

Another limitation of anti-GRP-R treatment is that it is not a targetable mutation, rather GRP-R is ubiquitously expressed with physiologic functions. However, GRP-R has been shown to be upregulated in neuroblastoma. In addition, clinical studies have demonstrated success with radioactive GRP-analogs and GRP-R antagonists for both diagnostic tumor imaging and radiotherapy of GRP-R-overexpressing cancers [37, 39–41].

The anti-GRP-R mAb is a potential novel anti-cancer immunotherapy that effectively inhibits neuroblastoma tumorigenesis via modulation of the PI3K/AKT pathway and induction of ADCC. Despite continued research advances, high-risk neuroblastoma remains difficult to treat. However, GRP-R mAbs have the potential to improve survival outcomes for patients with neuroblastoma through the use of a novel immunotherapeutic strategy while mitigating the morbidity of traditional anti-cancer treatments. Furthermore, combination treatment with blockade of the PD-L1/PD-1 immune checkpoint may sustain the activity of effector NK cells and enhance GRP-R mAb-mediated ADCC on neuroblastoma tumor cells.

## Supporting information

**S1 Fig. Determination of the binding affinity of GRP-R mAbs.** The $K_D$ value of GRP-R mAbs was measured using the 8-channel Octet RED96 system. The kinetic buffer without

antibody was used as correction. The *KD* values were obtained by fitting the data to a 1:1 binding model.
(TIF)

**S2 Fig. The specificity binding analysis of GRP-R mAbs.** GRP-R mAbs cross binding test using ELISA coated with four different GRP-R extracellular domains E1, E2, E3, and E4. GRP-R mAb-1, mAb-17, and mAb-21 have no cross binding with GRP-R extracellular domain E2, E3, and E4.
(TIF)

**S3 Fig. Immune checkpoint PD-L1 and PD-L2 expression in neuroblastoma BE(2)-C cells.** The mRNA level of PD-L1 and PD-L2 was measured by qRT-PCR and normalized to GAPDH in BE(2)-C cells treated with IFNγ at 10 ng/ml for 24 hours.
(TIF)

**S4 Fig. Immune checkpoint PD-L2 expression in neuroblastoma cells.** The mRNA level of PD-L2 was measured by qRT-PCR and normalized to GAPDH in five human neuroblastoma cell lines. All cells were treated with IFNγ at 0, 1, or 10 ng/ml for 24 hours.
(TIF)

**S1 Raw images.**
(PDF)

## Author Contributions

**Conceptualization:** Jingbo Qiao, Rachael A. Clark, Sora Lee, Ningyan Zhang, Dai H. Chung.

**Data curation:** Jingbo Qiao.

**Formal analysis:** Jingbo Qiao, Junquan Liu, Jillian C. Jacobson, Sora Lee.

**Funding acquisition:** Zhiqiang An, Dai H. Chung.

**Investigation:** Jingbo Qiao, Junquan Liu, Jillian C. Jacobson, Sora Lee, Li Liu.

**Methodology:** Jingbo Qiao, Junquan Liu, Rachael A. Clark, Sora Lee, Li Liu, Ningyan Zhang.

**Supervision:** Jingbo Qiao, Zhiqiang An, Ningyan Zhang, Dai H. Chung.

**Validation:** Jingbo Qiao.

**Writing – original draft:** Jingbo Qiao, Junquan Liu, Jillian C. Jacobson, Ningyan Zhang.

**Writing – review & editing:** Jillian C. Jacobson, Rachael A. Clark, Dai H. Chung.

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
