## [Decision Letter · Decision Letter 0]

26 Aug 2022

PONE-D-22-10312Anti-GRP-R monoclonal antibody antitumor therapy against neuroblastomaPLOS ONE

Dear Dr. Chung,

Thank you for submitting your manuscript to PLOS ONE. After careful consideration, we feel that it has merit but does not fully meet PLOS ONE’s publication criteria as it currently stands. Therefore, we invite you to submit a revised version of the manuscript that addresses the points raised during the review process.

Both reviewers have raised a number of minor issues which request you consider before the manuscript is accepted for publication.

We look forward to receiving your revised manuscript.

Kind regards,

Salvatore V Pizzo

Academic Editor

PLOS ONE

Journal Requirements:

3. To comply with PLOS ONE submissions requirements, in your Methods section, please provide additional information on the animal research and ensure you have included details on (1) methods of sacrifice, (2) methods of anesthesia and/or analgesia, (3) efforts to alleviate suffering, (4) tumor volume at sacrifice and (5) basic housing.

4. As part of your revision, please complete and submit a copy of the Full ARRIVE 2.0 Guidelines checklist, a document that aims to improve experimental reporting and reproducibility of animal studies for purposes of post-publication data analysis and reproducibility: https://arriveguidelines.org/sites/arrive/files/Author%20Checklist%20-%20Full.pdf (PDF). Please include your completed checklist as a Supporting Information file. Note that if your paper is accepted for publication, this checklist will be published as part of your article.

This work was supported by a grant from the National Institutes of Health (R01 DK61470). This work was supported in part by the Welch Foundation grant AU-0042-20030616 (ZA), and Cancer Prevention and Research Institute of Texas (CPRIT) Grants RP150230, RP150551, and RP190561 (ZA).

This work was supported by a grant from the National Institutes of Health (R01 DK61470). This work was supported in part by the Welch Foundation grant AU-0042-20030616 (ZA), and Cancer Prevention and Research Institute of Texas (CPRIT) Grants RP150230, RP150551, and RP190561 (ZA).

However, funding information should not appear in the Acknowledgments section or other areas of your manuscript. We will only publish funding information present in the Funding Statement section of the online submission form. 

This work was supported by a grant from the National Institutes of Health (R01 DK61470). This work was supported in part by the Welch Foundation grant AU-0042-20030616 (ZA), and Cancer Prevention and Research Institute of Texas (CPRIT) Grants RP150230, RP150551, and RP190561 (ZA).

7. We note that you have included the phrase “data not shown” in your manuscript. Unfortunately, this does not meet our data sharing requirements. PLOS does not permit references to inaccessible data. We require that authors provide all relevant data within the paper, Supporting Information files, or in an acceptable, public repository. Please add a citation to support this phrase or upload the data that corresponds with these findings to a stable repository (such as Figshare or Dryad) and provide and URLs, DOIs, or accession numbers that may be used to access these data. Or, if the data are not a core part of the research being presented in your study, we ask that you remove the phrase that refers to these data.

Reviewers' comments:

Reviewer's Responses to Questions

**Comments to the Author**

1. Is the manuscript technically sound, and do the data support the conclusions?

Reviewer #1: Yes

Reviewer #2: Yes

2. Has the statistical analysis been performed appropriately and rigorously? 

Reviewer #1: Yes

Reviewer #2: Yes

3. Have the authors made all data underlying the findings in their manuscript fully available?

Reviewer #1: Yes

Reviewer #2: Yes

4. Is the manuscript presented in an intelligible fashion and written in standard English?

Reviewer #1: Yes

Reviewer #2: Yes

5. Review Comments to the Author

Reviewer #1: 1. P 5, top: How does the statement about N-803 relate to this work?

2. P 14, middle: How many PDXs were evaluated? What was the breakdown of risk-category of the patients from which these PDXs were derived?

3. P 16, “This suggests that the immune checkpoint pathway…” What data presented demonstrates that activation of these immune checkpoint pathways suppressed the measured ADCC? If this is speculation, then this statement should be confined to the discussion.

4. P 18, bottom: Do these data that “there was no difference in tumors treated with GRP-R mAb alone” suggest the only mechanism of antitumor effects of this antibody are through ADCC? Please comment.

5. P 19, first paragraph: “highly effective treatment strategy”. Is this not somewhat of an overstatement? Please comment.

6. P 20, bottom: How did you demonstrate that the decrease in ADCC at 24 hours was a direct result of activation of the PD-1/PD-L1 pathway? Please clarify.

7. P 22, top and references 48 and 49: I agree that anti-GRP mAb would benefit from clinical testing but it is unclear to me how these two studies suggest using an anti-GRP mAb would be a “highly effective immunotherapeutic strategy”. Please comment.

8. P 23, last paragraph, also see comment #4 above: do these antibodies inhibit NB tumorigenesis? Please comment.

9. P 23, last sentence: The use of “will” seems too definitive for the data presented?

Reviewer #2: Qiao et al. established anti-GRP-R mAbs and mAb-1 blocked the GRP-ligand activation of GRP-R and the downstream signaling. Efficacious anti-tumor effects were then confirmed in vitro and in vivo. In this study, GRP-R mAb-1 can be a novel immunotherapy in the treatment of high-risk neuroblastoma patients. This manuscript can be acceptable with minor revision.

Major points

1. Current figures should be replaced with higher res ones.

2. The authors described that the affinity of antibodies was measured by Octet RED96. However, we did not find any data to support this.

Minor points

1. Consider not to use “the highest affinity” in the Abstract, detailed EC50/KD values was better. Similarly, “The cytotoxic effects of mAb-1 were confirmed in vivo using a murine tumor xenograft model”, please make a conclusion with detailed result.

2. The affinity of an antibody does not equal to EC50. It is said that “GRP-R mAb-1 has the highest affinity with the lowest EC50 at 4.607 ng/ml”, I recommend to mention something instead of it.

3. It is said that “Four peptides of the GRP-R extracellular domains (E1, E2, E3, and E4) were synthesized and used to screen the phage library for scFv binders” and only four mAbs against E1 were obtained. What’s the panning result of scFv binders against E2/E3/E4? Do the four antibodies have the cross reactivity to E2/E3/E4? And we do not see any control peptide/protein in Figure 1D.

4. Scale bar is missing in Figure 2E

5. What does abbreviation BBS stand for? It’s missing.

6. Some typos. For example, IFNg should be IFN-γ; “CD3-CD56+” , the “-” and “+” need to be superscripted; “Alex 488“ should be “Alexa Fluor 488”, et. al.

6. PLOS authors have the option to publish the peer review history of their article (what does this mean?). If published, this will include your full peer review and any attached files.

Reviewer #1: No

Reviewer #2: No

---

## [Author Response · Author response to Decision Letter 0]

18 Oct 2022

We have addressed all previously raised critiques and comments in the attached revised manuscript. We hope the Editorial Board will find our revised manuscript suitable for publication in PLOS ONE.

---

## [Decision Letter · Decision Letter 1]

8 Nov 2022

Anti-GRP-R monoclonal antibody antitumor therapy against neuroblastoma

PONE-D-22-10312R1

Dear Dr. Chung,

We’re pleased to inform you that your manuscript has been judged scientifically suitable for publication and will be formally accepted for publication once it meets all outstanding technical requirements.

Kind regards,

Salvatore V Pizzo

Academic Editor

PLOS ONE

Additional Editor Comments (optional):

Reviewers' comments:

Reviewer's Responses to Questions

**Comments to the Author**

1. If the authors have adequately addressed your comments raised in a previous round of review and you feel that this manuscript is now acceptable for publication, you may indicate that here to bypass the “Comments to the Author” section, enter your conflict of interest statement in the “Confidential to Editor” section, and submit your "Accept" recommendation.

Reviewer #2: All comments have been addressed

2. Is the manuscript technically sound, and do the data support the conclusions?

Reviewer #2: Yes

3. Has the statistical analysis been performed appropriately and rigorously? 

Reviewer #2: Yes

4. Have the authors made all data underlying the findings in their manuscript fully available?

Reviewer #2: Yes

5. Is the manuscript presented in an intelligible fashion and written in standard English?

Reviewer #2: Yes

6. Review Comments to the Author

Reviewer #2: (No Response)

7. PLOS authors have the option to publish the peer review history of their article (what does this mean?). If published, this will include your full peer review and any attached files.

Reviewer #2: No

---

## [Editor Report · Acceptance letter]

8 Dec 2022

PONE-D-22-10312R1 

Anti-GRP-R monoclonal antibody antitumor therapy against neuroblastoma 

Dear Dr. Chung:

I'm pleased to inform you that your manuscript has been deemed suitable for publication in PLOS ONE. Congratulations! Your manuscript is now with our production department. 

Kind regards, 

on behalf of

Dr. Salvatore V Pizzo 

Academic Editor

PLOS ONE